# Properties of Fiber-Reinforced One-Part Geopolymers: A Review

**DOI:** 10.3390/polym14163333

**Published:** 2022-08-16

**Authors:** Guoliang Guo, Chun Lv, Jie Liu, Li Wang

**Affiliations:** 1College of Architecture and Civil Engineering, Qiqihar University, Qiqihar 161006, China; 2College of Light-Industry and Textile Engineering, Qiqihar University, Qiqihar 161006, China; 3Engineering Research Center for Hemp and Product in Cold Region of Ministry of Education, Qiqihar 161006, China

**Keywords:** one-part geopolymer, fiber-reinforced, workability, mechanical property, durability

## Abstract

Geopolymers have the advantages of low carbon, being environmentally friendly and low price, which matches the development direction of building materials. Common geopolymer materials are also known as two-part geopolymers (TPGs). TPGs are usually prepared from two main substances, which are formed by polymerization of a silicoaluminate precursor and an alkaline activator solution. The TPG has many limitations in engineering application because of its preparation on the construction site, and the use of solid alkaline activator in one-part geopolymers (OPGs) overcomes this shortcoming. However, the brittleness of OPGs such as ceramics also hinders its popularization and application. The properties of the new OPG can be improved effectively by toughening and strengthening it with fibers. This review discusses the current studies of fiber-reinforced one-part geopolymers (FOPGs) in terms of raw precursors, activators, fibers, physical properties and curing mechanisms. In this paper, the effects of the commonly used reinforcement fibers, including polyvinyl alcohol (PVA) fiber, polypropylene (PP) fiber, polyethylene (PE) fiber, basalt fiber and other composite fibers, on the fresh-mixing properties and mechanical properties of the OPGs are summarized. The performance and toughening mechanism of FOPGs are summarized, and the workability, macroscopic mechanical properties and durability of FOPGs are investigated. Finally, the development and engineering application prospect of FOPGs are prospected.

## 1. Introduction

Geopolymer is a kind of inorganic polymer material formed by polycondensation of silicon-aluminum raw materials. It is a silicon-aluminum cementitious material produced by mixing low-calcium natural minerals or industrial wastes with an alkaline activator. It has a three-dimensional Al-Si network structure similar to that of natural zeolite. In the 1970s, French scientist Joesph Davidovits and his team prepared the earliest geopolymer materials by mixing alkaline solutions with calcined kaolin limestone and dolomite [1]. In recent years, due to a wide source of raw materials, simple process, low energy consumption, low environmental pollution and other excellent characteristics [2,3,4,5], the geopolymer has been regarded as a kind of environmentally friendly silicon-aluminum inorganic cementitious material, which has great development prospect and is expected to replace the Portland cement [6].

In fact, what is commonly referred to as alkali-excited binders is one of the geopolymer materials [7]. Similar to other geopolymers, it refers to inorganic binders produced by the reaction of precursor materials with potential hydraulic properties and alkaline activators [8,9]. Many studies show that alkali-activated cementitious materials have excellent physical and mechanical properties, such as fast hardening, early strength, corrosion resistance, freeze–thaw resistance and thermal stability. This geopolymer has been extensively studied by scientists and has been preliminarily applied in engineering [10].

Nowadays, to improve the toughness of geopolymers, scientists modified the matrix materials or added fibers to improve the brittleness of geopolymers. Commonly used reinforcement fibers include organic polymer fibers, cellulose fibers and other inorganic fibers. Organic fibers include polyvinyl alcohol fibers (PVA) [11], polyethylene fibers (PE), polypropylene fibers (PP) [12,13,14], plant straw fibers [15,16], etc. Inorganic fibers include basalt fibers, carbon fibers [17,18], steel fibers [19,20,21] and other types of fibers [22]. By toughening and strengthening the matrix of geopolymers with fibers, the optimization level of various fibers on geopolymers was explored, including fluidity, condensation, hardening, compressive strength, flexural strength and toughness [23,24].

The one-part geopolymers (OPGs) reviewed in this paper were usually based on fly ash, slag, kaolin, and other waste materials were also used [25,26,27]. FOPGs were synthesized by a solid state activator and toughened by fibers [28]. Usually, solid alkaline materials, such as Na_2_SiO_3_, were used as the activator. Through exploring the different material ratios, activator content, fiber type and content in the precursor, the influence rules of the workability, mechanical properties and durability of FOPGs were analyzed. Comprehensive studies showed that toughening and strengthening OPGs with fibers can effectively improve their macroscopic mechanical properties, including flow characteristics, compressive and flexural strength [29]. The microstructure of FOPG was characterized by means of microscopic analysis, and the stress transfer between the fibers and the matrix was explained in depth by means of microstructure analysis and comparative analysis of macroscopic mechanical properties. At the same time, the modification mechanism of fibers was explored to provide the basis for the structural design and engineering application of FOPGs.

## 2. Two-Part Geopolymer (TPG) and One-Part Geopolymer (OPG)

The main difference between OPG and TPG is the difference in the activator. The activator of OPG is a solid powder, and the activator of TPG is a liquid solution. The typical preparation process of traditional TPGs is relatively clear. First, silicon-aluminum raw materials are activated by calcination, and then, soluble alkali silicon-aluminate solution is used to dissolve and excite. Finally, after forming the geopolymer component, it is cured at 20–120 °C [30]. TPGs are prepared by reactive polymerization of silicaluminate precursors and alkaline activator solutions, such as sodium hydroxide (NaOH), potassium hydroxide (KOH) and sodium silicate, or their combinations.

Because traditional TPGs are prepared on construction sites, there are many limitations in engineering applications [31,32]. First of all, the synthesis and preparation of geopolymers are carried out on site, which will cause inconvenience in construction. Secondly, the alkaline solution is highly corrosive, and the construction process must be carried out under perfect protective measures, which hinders the large-scale use of geopolymers [33]. In addition, it is difficult to control the quality of different liquid activators, which reduces the efficiency of mass production of geopolymers.

As activators directly affect the hydration development process of geopolymers and the structure of hydration products, different activators have great influence on the properties of geopolymers [34,35]. The commonly used activator types mainly include: basic metal oxides or hydroxide, alkaline earth metal oxides or their hydroxides; non-silicate system weak acid salts, alkali silicate; alkali aluminate, basic aluminosilicate, non-silicate system strong acid salts, etc. Among these types, common activators include caustic soda, sodium silicate, sodium carbonate, sodium sulfate, lime, etc. [36,37].

Currently, the most studied types of geopolymer activators are caustic soda and sodium silicate [38]. Alkaline-activated cementitious materials with good performance can be obtained by using sodium silicate excitation, and their finished products have a high strength [39,40] and excellent resistance to chemical erosion [41,42]. Compared with sodium silicate excitation, the strength of geopolymers obtained by caustic soda excitation is relatively lower [43,44], and its porosity is relatively larger [45,46,47]. Engineering practice shows that the high alkalinity of the two activators brings risks to construction, such as high price of the activator [48], easy shrinkage and cracking [49], fast setting time and poor fluidity of the mixture [50,51]. With that in mind, some scientists are starting to look at other kinds of activators. Rashad et al. [52] used a sodium sulfate activator to modify the slag-based precursor with traditional Portland cement, which significantly improved the strength of geopolymers. Yuan et al. [53] combined sodium silicate, sodium carbonate and sodium hydroxide as the activator and tested the strength and hydration products of alkali slag cement, indicating that the content of sodium silicate determines the strength of the cementing material, while sodium carbonate mainly affects the gel structure of hydration products.

At present, the research of geopolymers mainly focuses on the modification of TPGs. Using fibers to enhance the toughness of the geopolymer is an important research content in this field [54,55,56]. Based on the current research status of geopolymers, some researchers put forward the concept of OPGs. However, there are relatively few studies on OPGs, especially on fiber-reinforced one-part geopolymers (FOPGs) [57]. The matrix of FOPG mainly consists of precursors, activators and other auxiliary materials. Similar to the TPG, the precursor materials are mostly silicon aluminate materials, such as kaolin, fly ash and slag. However, the activator in OPG is different from TPG, which uses a solid activator. OPGs are usually synthesized by the solid Na_2_SiO_3_ activator.

Specifically, an OPG is a mixture of a solid alkaline activator and a precursor of aluminum silicate that can be directly mixed with water, similar to the traditional cement concrete [58]. The early studies on OPGs mainly focused on the calcination of silicoaluminate with solid alkali metal hydroxide or carbonate, and the clinker was crushed into clinker and then added into water for preparation [59]. In addition, similar to TPGs, OPGs also have the shortcomings of insufficient toughness and brittle failure, so using fibers as the toughening materials is an inevitable choice [60,61].

In fact, from the meaning of OPG, we naturally come to the concept of FOPG. FOPG is a material prepared by composite preparation of silicon aluminate precursor and solid basic activator with fibers. According to engineering needs, different types, aspect ratio and content of fibers can be selected. Through the composite of fibers and OPG, the toughness and other mechanical properties of FOPGs can be effectively improved.

## 3. Properties of Different Fibers

The types of fibers are closely related to the mechanical properties, geometric properties and elongation of fibers and have great influence on the properties of FOPGs. Reinforced fibers commonly used in FOPGs include organic polymer fibers, cellulose fibers, steel fibers, carbon fibers and basalt fibers [62,63]. These properties are shown in Table 1. It can be seen that the properties of different types of fiber are greatly different, and the enhancement effect of geopolymers is also different. Organic polymer fibers have high tensile strength and good properties, which are similar to traditional inorganic fibers. The elastic modulus and tensile strength of cellulose fibers are low, which mainly improve the toughness and impact resistance of FOPGs.

In engineering practice, these characteristics of different types of fiber should be fully utilized. According to the needs, FOPGs were prepared by mixing fibers in OPGs. Nematollahi et al. studied PE and PVA fiber-reinforced OPGs prepared with fly ash and blast furnace slag as precursors and lime as the activator [71,72,73]. In these experimental studies, the fiber addition is usually 2% by volume. Relevant studies have shown that FOPGs with excellent performance can be successfully developed by a proper combination of fiber, precursor and solid activator.

## 4. Properties of FOPGs

As an inorganic cementitious material, OPGs exhibit high ceramic-like brittleness and relatively low fracture energy [74]. To overcome this weakness, combining fibers with the geopolymer matrix is an effective technique to improve the flexural properties of composites [75,76]. Of course, it also has varying degrees of influence on working performance and other performances. See Table 2 for the related research on the mix composition of TOPGs.

As can be seen from Table 2, slag or a combination of slag and fly ash are mostly used as precursors of TOPGs. The slag precursor is prone to crack due to its large shrinkage, which can be improved by adding fly ash. Most of the activators are anhydrous sodium silicate powder. Traditional steel fiber is used more often; more choose polymer fiber and basalt fiber, but cellulose fiber is also chosen. Curing is mainly ambient temperature air curing, water tank curing and heat air curing. The water solid ratio is between 0.3 and 0.45, and the fine aggregate usually adopts standard sand, but it also adopts fine sand or no fine aggregate.

### 4.1. Workability of FOPGs

As can be seen from the previous discussion, the content of the activator has a great influence on the fluidity of fresh FOPGs [79]. On the one hand, as the catalyst of alkali excited reaction, the activator can effectively stimulate the activity of the precursor and catalyze the polymerization reaction. On the other hand, it is also an important participant in the reaction, interacting with Ca^2+^ to produce C-S-H gels. Therefore, the content of the activator is an important factor affecting the property of geopolymers [80]. The solid Na_2_SiO_3_ activator was used, and the effects of different contents of the activator on the properties of the polymer were different. As can be seen from the influence of the content of the activator on the fluidity and setting time of the geopolymer, the fluidity of FOPGs basically decreases with the increase in Na_2_SiO_3_ content. At the same time, the increase in alkali content of the geopolymer accelerates the process of hydration reaction and promotes the polymerization reaction.

In recent years, FOPGs have attracted widespread attention from researchers, and the research results are gradually increasing. Panda et al. [81] conducted an experimental study on 3D printed geopolymers and found that the TPGs had high viscosity due to the high content of the alkaline activator solution. By using solid aluminosilicate precursors and solid activators, a solution was provided for the synthesis of OPGs. In this work, the OPG mixture has good flow characteristics and thixotropy.

Shah et al. [66] studied the properties of micro-steel fiber, PVA and basalt fiber reinforced FOPGs. The fluidity, compressive strength, flexural strength and splitting tensile strength of composites were evaluated according to the influence of different fiber types and contents. The results showed that the mechanical properties of FOPGs containing three kinds of fibers were improved. The reinforcement effect of micro-steel fibers was best under the tensile load. The addition of PVA fibers could improve the deformation ability and the strength of the mixture. Basalt fibers also produced positive results for OPGs. The effect of different fibers on the flow of FOPGs was slightly different. Compared with non-fiber mortar, the flow diameter of non-fiber mortar was the largest, and the addition of three kinds of fiber decreased the flow diameter. Among the different fibers, PVA fibers reinforced geopolymer had the lowest workability, followed by basalt fibers and micro-steel fibers. Different researchers have also reported flow losses of different fibers added to FOPGs [82]. The unstable flow of the paste due to the addition of fibers to the composite resulted in a loss of workability. The basalt fiber and micro-steel fiber reinforced FOPGs remained workable up to 2% fiber volume fraction, while PVA remained workable up to 1.5%.

The relationship between the relative slump of FOPGs with different fiber contents was shown in Figure 1.

The groups in the figure show the mixture ratio and fiber distribution of the precursor slag and fly ash used in each series. P:EGC-S represents the slag base geopolymer (100% Slag), and considering the effect of sand addition to the mixture, P series represents fine sand without drying treatment. M:EGC-FA/S stands for the mixed base geopolymer (50% Fly ash and 50% Slag). The fine sand used in the M series was dried in an oven. Each series consisted of five different ST and PE fiber blends while maintaining a total fiber volume fraction of 2%. It can be seen from the figure that the slump of the mixed base geopolymer (P:EGCFA/S) is slightly higher than that of the slag base geopolymer (P:EGC-S), which is due to the ball-bearing effect of FA particles. The relative slump of the composite decreases after the fiber is added to the matrix. Apparently, this decrease is more pronounced when higher PE volume fractions are used. In other words, the greater the amount of steel fibers used in the matrix, the higher the workability. This trend is reflected in both the mixed and slag-type geopolymers. According to the observation of all geopolymer mixtures in the study, the mixture has good workability and uniform fiber dispersion in the process of mixing and compaction [83].

With the addition of single fibers, the fluidity of FOPGs decreased with the increase in fiber content, and it showed higher consistency than that of the geopolymer mortar without fiber. This indicates that the addition of fibers in FOPGs will significantly affect their workability, and fibers were well dispersed in the geopolymer slurry without agglomeration [84]. For organic polymer fibers, both PVA and PE fibers can obviously reduce the fluidity of the mortar. Because the PVA and PE fibers show high cohesion in water, the fibers evenly dispersed in the geopolymer gel play a supporting role, so that the fiber and the geopolymer matrix are closely combined. At the same time, with the increase in fiber content, the amount of coated geopolymer slurry increases, which prevents the flow diameter of the mortar body from further expanding. This shows that the flow characteristics are closely related to the fiber volume content in FOPGs.

### 4.2. Mechanical Properties of FOPGs

Similar to other concrete composites, the mechanical properties of FOPGs mainly include the compressive strength, flexural strength, tensile strength and tensile toughness.

#### 4.2.1. Compressive Properties of FOPGs

Generally, the compressive strength of FOPGs decreases with the increase in fiber content, regardless of the kind of fiber added. Compared with the failure mode of OPGs without fiber addition, only matrix cracking occurred in FOPGs, but no fragmentation failure occurred.

Perumal’s team studied two different FOPGs [77]. Two types of precursors—pure slag and ternary mixed slag—were used to understand their efficiency in FOPGs. These matrices were reinforced with three different fibers (steel, glass and basalt) of 1% volume fraction to improve their flexural properties. The results showed that the compressive strength of steel fiber was better than that of mineral fiber.

Sood et al. studied binary and ternary combinations of geopolymer precursors, and the activators used two kinds of reagents in the form of powder [78]. Sixteen different combinations of SiO_2_/Al_2_O_3_, Na_2_O/SiO_2_, CaO/SiO_2_ and Na_2_O/Al_2_O_3_ and PVA fibers were formed to study their fresh state and hardening properties. The results showed that the compressive strength of the 56-day FOPG was 54 MPa, which was the best binder, with the ternary precursor combination of C-grade fly ash 25%, F-grade fly ash 35%, slag fine particles 40% and the activator combination of calcium hydroxide and sodium metasilicate ratio 1:5.5. The addition of 2% PVA fibers to the FOPG had no significant effect on its compressive strength. However, it contributed to the relaxation of the matrix shrinkage strain through micro-constraint.

Alrefaei et al. [69] studied different hybrid combinations of steel fibers and PE fibers. Two FOPGs were prepared under the condition where the total fiber volume was 2%. One was 100% slag-activated synthesis, and the other was a mixture of 50% fly ash and 50% slag. The effects of different precursor materials and fiber contents on the properties of FOPGs were studied. It was found that the two FOPGs had comparable compressive strength. Compared with the mixed base FOPG, the slag base FOPG had a relatively better tensile response. The results showed that the matrix density of slag FOPG is higher than that of mixed FOPG. Another study quantitatively evaluated the effects of curing conditions (heat curing and room temperature curing) and fiber type on the macroscopic properties of the matrix and composites [85,86]. The matrix fracture properties and fiber–matrix interface properties were determined by a fracture toughness test and a single fiber drawing test, respectively. The fiber bridging constitutive relationship of composites was calculated by the micromechanical model, and the microstructure of the composites was related to the macroscopic tensile properties of the composites. The results showed that curing at room temperature increased the compressive strength and tensile strength of the PP FOPGs but decreased their tensile ductility. Compared with the PVA FOPGs, the PP FOPGs had a lower compressive strength and tensile strength but higher tensile ductility.

Nematollahi et al. also found that the addition of 2% PVA fibers to FOPGs had no significant effect on the compressive strength [67]. Bernal et al. [87] found that the addition of fibers reduced the compressive strength of the geopolymer. However, with the increase in fiber volume, the splitting tensile strength and flexural strength were greatly improved, which were 3.75–4.64 MPa and 6.40–8.86 MPa, respectively, after curing for 28 d. The addition of steel fiber improved the properties related to durability, such as water absorption and water permeability. At the same time, different fiber contents may reduce the flexural strength and compressive strength of FOPGs [88,89].

OPGs have good compressive strength and durability. However, their low flexural strength, tensile strength, strain capacity and brittleness limit their application in some cases. Shah [66] studied the properties of a one-part alkali-activated fly ash slag mortar reinforced with micro-steel, PVA and basalt fibers. The effects of fiber type and content on the mortar were evaluated from the slump, water absorption, compressive strength, flexural strength, splitting tensile strength and load deflection curves under flexural and splitting tensile strength. The results showed that the mechanical properties of one-part alkali-active mortar could be improved by adding three kinds of fibers. Under the tensile load, micro-steel fibers were most beneficial to improve the displacement capacity of the mixture, and the addition of PVA fibers significantly improved the strength performance of the mixture. Basalt fibers also achieved positive results in mechanical properties, showing good potential for enhancing the mechanical properties in FOPGs.

In engineering practice, fibers should be correctly added to FOPGs according to the engineering needs. Generally, the addition of a fiber has no significant effect on the compressive strength of composites. In particular, the addition of excessive fibers to the composite may reduce the compressive strength of the FOPGs.

#### 4.2.2. Flexural Strength of FOPGs

The main purpose of adding fibers into cement mortar or concrete is to optimize the flexural strength of the hardened matrix and transform it from brittle fracture mode to ductile fracture mode. The flexural strength of the OPG matrix increases obviously with fiber incorporation. When the fiber content is less than 2%, the flexural strength of FOPGs at each age increases with the increase in fiber volume content, especially the early flexural strength. When the fiber content continues to increase, due to the restriction of the forming process, the fiber aggregation phenomenon occurs, and the matrix is weak in the zone with less fiber distribution. During the experiment, it was also observed that with the increase in fiber volume content, the matrix showed the failure mode of multiple cracks. These results indicate that PVA fibers contribute to the improvement of the flexural strength of the geopolymer matrix, especially in the early flexural strength. Natali et al. explored the influence of carbon fiber and PVC fibers on the properties of metakaolin slag-based geopolymers. They found through experiments that all samples with fiber added showed increased flexural strength, and this improvement effect was particularly obvious when carbon fibers and PVA fibers were added. At the same time, the ductility of the matrix was improved after cracking, which was consistent with the results of this test [90].

In the FOPGs system, the fiber length-diameter ratio, volume content and stirring mode have significant effects on the mechanical properties of the matrix, so the selection of the preparation process is particularly important. Table 3 shows the test results of a typical mix of FOPGs [67].

In Table 3, all numbers are mass ratios of the precursor weight (fly ash + slag), except fiber content (volume fraction). All fiber contents are 2%. A and H are different curing conditions (A—Ambient Temperature Air, H—Heat Air). It can be seen that different kinds of fibers and curing conditions have different effects on the properties of composites.

Figure 2 shows the tensile stress–strain response curves of FOPGs. It can be seen that regardless of the curing conditions and fiber types, they all exhibit strong strain hardening behavior. Different from the PE fibers (Figure 2a,b), PVA fibers (Figure 2c) form chemical bonds with the matrix (PVA-A) due to their hydrophilicity, and their initial crack strength is higher. The results show that the FOPG of PE (PE-H, PE-A) had moderate to high tensile strength (3.3–4.2 MPa) and very high tensile strength (4.9–5.5%), while the PVA-A had high tensile strength and very high tensile resistance—4.6 MPa and 4.2%, respectively.

As mentioned above, the fiber plays a positive role in the improvement of the flexural strength of FOPGs. The fiber effectively improves the cracking resistance of the matrix and greatly improves the flexural strength of FOPGs. The improvement difference of the flexural strength mainly depends on the mechanical properties of the fibers themselves and the dispersion degree of the fibers in the geopolymer matrix. When the fiber volume content is more than 2%, the fiber strength decreases due to uneven dispersion. Therefore, it is considered that 2% is the optimal proportion to improve the flexural strength of FOPGs. In addition, it was found that when the composite fiber was added, the flexural strength of the FOPG was not significantly increased, which was basically the same as that of the single fiber.

Alrefaei et al. [68] tested the elastic modulus of FOPGs. The results showed that the compressive strength of the mixture of fly ash and slag was higher than that of slag, but the elastic modulus of slag was larger. The addition of fiber increased the elastic modulus of geopolymer composites compared with the non-fiber composites, and the increase in elastic modulus was related to the volume of steel fibers. In addition, all composites exhibited flexural hardening behavior, which was directly related to the volume of PE fibers contained in the composites. In general, the slag base exhibited better bending behavior than the mixed base.

Shah et al. [66] found that fiber type and content had a significant influence on the flexural properties of FOPGs. The load displacement curves under the flexural loading of FOPGs reinforced by 2% PVA, basalt and micro-steel fibers are shown in Figure 3. As can be seen from the figure, the geopolymer with no fibers suddenly cracked under peak load, presenting a typical brittle failure. The peak load and the corresponding displacement of the mortar were increased compared with those of the control mortar. The addition of 2% micro-steel fibers showed displacement hardening, and the other components showed displacement softening. Perumal et al. [77] found that the fracture energy of SF reinforced composites is about 4 times higher than that of mineral fibers reinforced geopolymers. In addition, regardless of the fiber type and properties, the flexural properties of ternary composites were higher than those of slag-based composites.

Figure 4 showed the flexural strength–deflection curve of polypropylene fiber (PPF) geopolymer after 56 days [88]. Similar to FOPGs, the results showed that the hardened geopolymer matrix without PPF had a high brittleness, and the strength of the first crack was higher than that of the matrix with fiber. On the contrary, the addition of PPF fibers increased the shrinkage defects and extra porosity of the matrix, resulting in the reduction in the effective cross section. Therefore, the flexural strength of the matrix itself was weakened, and the stress when the first microcrack appeared in the FOPG was reduced. A similar behavior was observed in other cementitious composites, where an increase in porosity led to a decrease in bending strength.

It was found that the fiber could improve the physical properties of FOPGs, and the fiber type and curing condition had a certain effect on the mechanical properties. Numerical calculation results showed that the flexural capacity of FOPGs was within the range of 10–40 kN/m^2^, and the allowable load was far lower than the ultimate capacity [91].

#### 4.2.3. Adhesion Mechanism between Fibers and Geopolymer Matrix

Similar to other fiber-reinforced geopolymer or concrete, the fiber toughening mechanism for FOPGs mainly includes fiber fracture, fiber pulling out, fiber debonding, crack bridging, crack deflection [92].

Fiber fracture consists of an effective breakage of the fiber during crack propagation [93]. Fiber pulling out means that the fiber at the crack tip slides out along the interface between the fibers and the matrix under the action of external tensile stress. Compared with fiber fracture, the toughening effect of fiber debonding is better. Fiber bridging refers to the difficulty of deflecting fibers with cracks facing specific distribution and direction. Crack deflection means that the crack tip causes the crack to bend and extend, which leads to the reduction in matrix stress and hinders crack growth. Generally speaking, the toughening effect is related to the volume ratio and aspect ratio of the reinforcement. The larger the volume ratio and aspect ratio, the better the toughening effect of crack bending.

As mentioned above, fiber plays a positive role in improving the flexural strength of FOPGs. The addition of fibers improves the properties of OPGs through toughening, strengthening and cracking resistance. Some scientists tested the microscopic morphology of FOPGs to further explain the improved mechanical properties of FOPGs from a microscopic perspective.

Fibers embedded in the matrix and the failure mode of the fiber are shown in Figure 5.

From these images, it can be seen that the microstructure of the steel fiber reinforced matrix is shown in Figure 5a. The matrix demonstrated debonding failure, and the matrix had a certain pull-out effect. The residual bond between the mortar and the fibers indicated the effective bond between the fibers and the mortar. In terms of fiber types, steel fiber was superior to glass fiber and basalt fiber in the contribution to the compressive strength of the FOPG. It could be observed that glass fiber clusters (see Figure 5b) and basalt fiber clusters (see Figure 5c) were still bound in the binder matrix, which explained the inefficient fiber distribution. In addition, basalt is unstable in highly alkaline environments and can cause fiber degradation [94].

In Figure 5c, the image of basalt fibers shows its reactivity with the matrix, and fibers are captured into the reaction product. This explains the strength loss caused by basalt fibers in FOPG systems and promotes the need for alkali-resistant coatings [95].

Abdollahnejad’s team also found similar SEM patterns in their study [65]. PVA and basalt fibers provided a good interfacial transition zone bonding between the fibers and the matrix, which was already attached to the fiber surface. Due to the smooth surface of the PP fibers, these fibers had weak bonding properties at the fiber/matrix, and therefore, the PP fibers were de-bonded to the surrounding matrix. In addition, one end of the PVA fibers was obviously fractured, while the other end was embedded in the hydration product of the matrix, indicating that when the main crack appeared, the PVA fiber continued to function under the external bending load until it was damaged by tension.

Shah et al. also studied the microstructure of FOPGs using SEM, as shown in Figure 6 [66]. Firstly, it can be seen from Figure 6a that the amorphous hydration products with apparent density and uniformity were formed, and the C-S-H and N-A-S-H gels generated by the reaction covered the surface of the geopolymer matrix. At the same time, the hydration reaction of the matrix occurred rapidly, and a large amount of heat was released, which caused the shrinkage of the matrix. The microcracks in the interfacial transition zone and in the matrix were generated. Figure 6b shows that hydration particles were attached to the surface of PVA fibers, and fibers were wrapped by the geopolymer slurry in good shape. This shows that the adhesion between PVA fibers and the matrix was good. The good adhesion of PVA fibers was the main reason that PVA fibers acted as bridges under external load and thus improved the strength of the mortar.

As for basalt fibers reinforced OPG, due to the poor adhesion between basalt fibers and the geopolymer matrix, a few fibers slipped, as shown in Figure 6c. Basalt fiber has a lower splitting tensile strength and flexural strength compared with PVA fiber, which is related to the slip of basalt fiber under tensile load. The slip of the basalt fiber may be caused by the smooth surface of the basalt fiber and the weak bond between the matrix and fibers. In terms of the failure mode, a path is formed at the interface between the matrix and the fibers, which is caused by the slip of the fiber in the matrix under tensile stress. In this process, the work carried out by external force is used to balance the breaking energy, which belongs to fiber pulling out.

The microscopic cracks formed by the micro-steel fibers can be observed in Figure 6d. When the fiber content was greater than the optimal value, the microcracks had negative effects on the mechanical strength of the composites. When the amount of micro-steel fiber was 2%, the mechanical strength decreased slightly because the formation of the microcracks had a negative effect on the amount of micro-steel fiber.

According to the test results, Alrfaei et al. plotted the relationship between the average fracture spacing and the fiber volume fractions [69], as shown in Figure 7. The crack spacing of the slag base geopolymer hybrid fibers composite was 1.6–5.1 mm, and that of the fly ash mixed with slag base geopolymer hybrid fibers composite was 1.9–9.3 mm. For the single PE fibers composite, the crack spacing of the fly ash-slag mixed base geopolymer was 19% larger than that of the slag base geopolymer, while for the hybrid fibers composite, the crack spacing of the fly ash-slag mixed base geopolymer was 80–150% larger than that of the slag base geopolymer. Therefore, the crossbreeding of steel fibers and PE fibers had better performance in the slag base geopolymer. As can be seen from the figure, with the increase in PE fiber addition, the crack spacing of the fly ash and slag mixed base geopolymer and the slag base geopolymer decreased. As mentioned above, the PE volume fraction has a direct influence on the multiple cracking behavior of the composites, and hence, on crack spacing. In addition, the slag base geopolymer has better cracking performance than the fly ash mixed with slag base geopolymer, so the crack spacing is relatively small.

### 4.3. Durability of FOPGs

Durability is related to the long-term performance of the composites and has had great importance attached to it by relevant researchers [96,97,98,99]. Abdollahnejad et al. [64] studied the effects of different types of fibers (steel, PVA, basalt and cellulose) and fiber combinations (single and mixed) on the mechanical properties and durability of FOPGs. All fibers were added at a volume fraction of 1%. The durability of FOPGs was studied by the properties of water absorption, acid resistance, high temperature resistance, carbonation resistance and freeze–thaw resistance. The experimental results showed that different types and combinations had great influence on the mechanical properties and durability of FOPGs. In addition, the influence of different types and combinations on high temperature resistance and freeze–thaw resistance was greater than that on acid resistance and carbonization resistance [100]. Coppola et al. [101,102] evaluated the durability of OPGs in different corrosive environments, such as calcium chloride and magnesium sulfate solutions, and compared them with conventional mortars. The experimental results showed that the alkali content was a key parameter for durability; the higher the alkali content, the higher the resistance under harsh conditions. In particular, the high-alkali-content geopolymer had a similar freeze–thaw resistance to the blast furnace slag cement mixture but lower than the Portland cement mortar, while the low activator geopolymer had very limited freeze–thaw resistance in a cold environment. FOPGs also had good acid resistance, but they were very susceptible to magnesium sulfate [103].

The use of hazardous alkali solutions is avoided in FOPGs, making these materials easier to handle and transport. The blast furnace slag can be used as a one-part alkali active binder and can achieve high compressive strength in the early stage. However, the use of slag alone has some disadvantages, including fast setting time and high drying shrinkage, which narrow the application range of these mixed components and lead to the formation of cracks separately, thereby shortening the service life. Fly ash can partially replace slag as a one-part alkali-activated material, which can not only reduce drying shrinkage but also prolong setting time. The effect of fly ash replacing more than 80% of slag on the fresh hardening performance of one-part alkali-active mortar was studied [104]. These properties were characterized by initial and final setting times, strength development and drying shrinkage rates. The one-part alkali-activated slag/fly ash mortar was reinforced with different fiber contents and combinations in order to improve its strength, further reduce the drying shrinkage rate and improve its freeze–thaw resistance. The results showed that adding fly ash and fiber could reduce the drying shrinkage and prolong the curing time. The mechanical properties of the slag/fly ash composite were weakened by the decrease in the slag content, so that the slag/fly ash composite could be fully considered in an engineering application. After drying and shrinking, geopolymer materials easily produce cracks, which is also an important index affecting their durability. The effects of different fiber types and combinations on the drying and shrinkage of FOPGs are shown in Figure 8 [65].

The addition of hybrid fibers increased the drying shrinkage more than that of single fibers. The only effective fiber that reduced the drying shrinkage was glass fiber with a 1.5 volume content, which reduced the final drying shrinkage by approximately 15%. Although the addition of all hybrid composite fibers increased the drying shrinkage rate, the increase was the lowest in 0.5Ba1PP, which was about 10%. Although the drying shrinkage rate of most reinforced mixtures was increased compared to the traditional reference mixture, the final drying shrinkage rate recorded was even lower than that of some traditional mixtures. The drying shrinkage of FOPGs was much lower than the crack value observed in other geopolymer substrates.

The compressive strength of the reinforced mixture before and after freeze–thaw cycles is shown in Figure 9. It can be seen that after 120 freeze–thaw cycles, the compressive strength of all the mixed materials decreased. According to the results, the compressive strength of the common mixture was reduced by about 10%, respectively, and the minimum and maximum compressive strength losses of 1.5 PVA and 1.5 PP were about 10% when reinforced by single fibers. In addition, for hybrid fibers reinforced materials, the maximum and minimum compressive strength decreased by about 50 and 15% at 0.75PVA0.75pp and 0.5PVA1Ba at 90 days, respectively.

In general, the workability of a FOPG decreases with the addition of fibers, which has an adverse effect on the compressive strength of the matrix, and its durability decreases to a certain extent. However, its flexural strength is greatly improved. On the one hand, the elastic modulus of the fiber is much lower than that of the geopolymer, and the matrix is limited by the bearing capacity of the fiber, which is easier to destroy than the mortar without fiber. On the other hand, the strength decline is related to the pores introduced into the mortar during the formation of the fiber-reinforced geopolymer sample. In addition, there is a porous weak layer in the interfacial transition zone between the polymer matrix and the fiber, which is also the reason why the presence of fiber can weaken the compressive strength of the geopolymer. At the same time, the stress distribution of fibers in the matrix changes under the loading. The failure mode of FOPGs also changes, and the fracture mode of OPGs changes from brittle fracture to plastic fracture. The FOPG exhibits more crack propagation and strain hardening characteristics, which improves the bending toughness of FOPGs.

## 5. Future Perspectives

From the perspective of the engineering application of green materials, although FOPGs have so many advantages, they still need to be improved and perfected in many aspects. However, the study of FOPGs still has some limitations. There are relatively few types of binder, which limits their engineering application. At the same time, there is a lack of research and application on the recycled aggregate, and few studies on straw fibers reinforced OPGs.

FOPGs consist mainly of fibers, precursors, activators and a perfect combination of these factors. Compared with the traditional steel fibers, basalt, carbon, polyvinyl alcohol, polyethylene, polypropylene fibers have unique characteristics in mechanical properties, scale dimension, cluster effect and morphological effect. Therefore, synthetic fiber reinforced and toughened cementitious composites have received great attention. Compared with fiber-reinforced cementitious composites, the research of FOPGs is relatively lagging behind. At the same time, in view of the differences in the physical and mechanical properties of different fibers, the distribution characteristics of fibers in reinforcement are obviously different, and the strengthening effect and action mechanism need to be further studied. Using slag alone as a precursor has the disadvantages of short solidification time, large drying shrinkage and the formation of cracks, so as to shorten the service life. Fly ash is used in FOPGs to partially replace slag, which can not only reduce the drying shrinkage but also prolong the solidification time. FOPGs use a solid activator; its variety is relatively limited, and the cost is high. From the previous study, it can be seen that the solid activator is mainly anhydrous sodium metasilicate (Na_2_SiO_3_). In the future, it is necessary to develop more varieties and lower the prices of activators in order for them to be more suitable for engineering applications.

## 6. Conclusions

Compared with TPGs, OPG is more suitable for engineering popularization and application. However, the brittleness of its ceramic-like materials hinders its further application in engineering. The mechanical properties can be improved by adding fibers. The commonly used fibers include PVA, PE, basalt and composite fibers, which have great influence on the fresh mixing performance and mechanical properties of slag-/fly-ash-based FOPGs. When fibers are added into FOPGs, the fluidity decreases greatly with the increase in fiber volume content. Because these fibers show good cohesion when they come into contact with water, the fibers are closely combined with the matrix. At the same time, with the increase in fiber content, the amount of wrapped geopolymer slurry increases, thus preventing the flow diameter of the mortar body from further expanding. The compressive strength of FOPG decreases with the increase in fiber volume content, while the flexural strength increases first and then decreases. The modification effect of fibers in the FOPG is mainly visible after the initial cracking of the matrix and before the failure. Under the action of external load, the excessive development of cracks can be inhibited by means of fiber pulling out and fiber fracture, and the energy consumption of external force destruction can be increased to improve the flexural toughness of FOPGs. Fibers have the effect of enhancing the toughening and strengthening of FOPGs. The energy consumption modes of fibers pull-out work and fracture work can improve the cracking resistance and bending resistance of FOPGs.

## Figures and Tables

**Figure 1 polymers-14-03333-f001:**
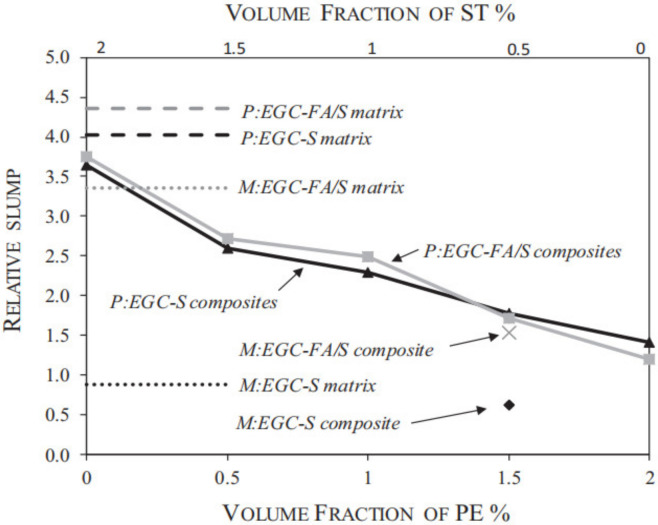
Relative slump versus PE and ST fiber volume fractions, Adapted with permission from [69], copyright 2018 Elsevier.

**Figure 2 polymers-14-03333-f002:**
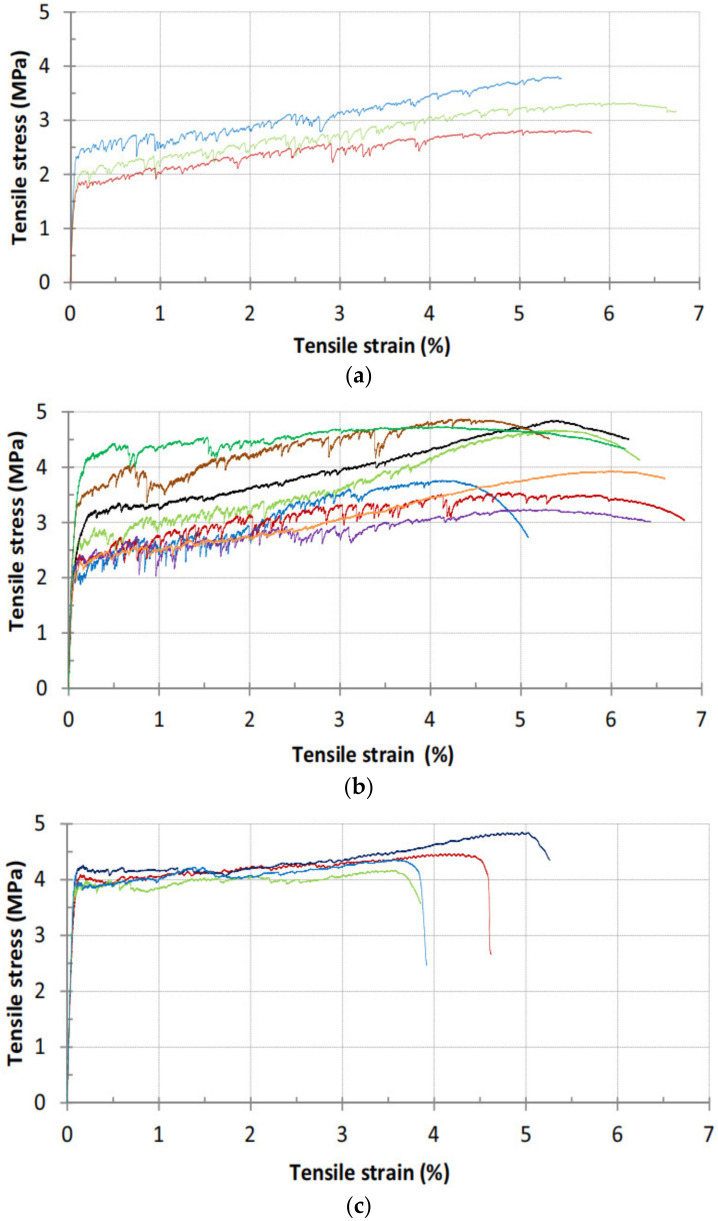
Tensile stress–strain responses of FOPG [67]. (**a**) Heat-cured FOPG (PE-H), (**b**) Ambient-temperature-cured FOPG (PE-A), (**c**) Ambient-temperature-cured FOPG (PVA-A). Adapted with permission from [67], copyright 2017 Elsevier.

**Figure 3 polymers-14-03333-f003:**
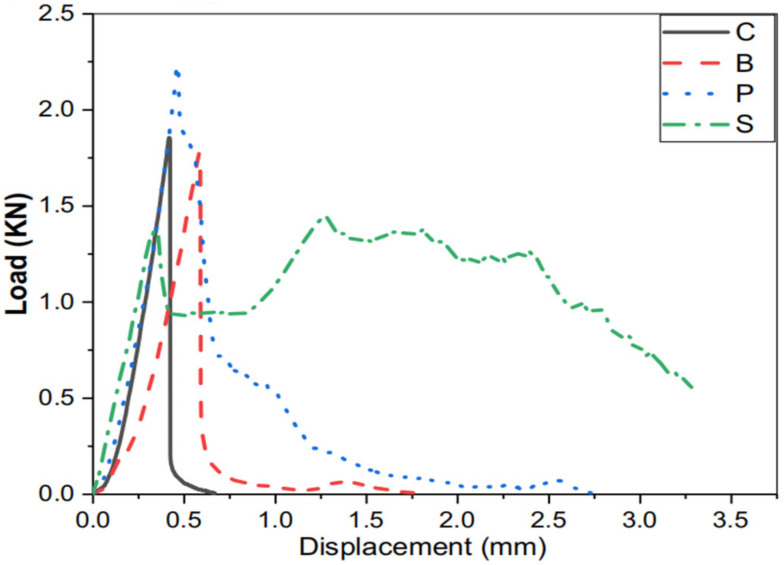
Load displacement curves under flexural loading. PVA fibers (P), basalt fibers (B), micro-steel fibers (S) and control (C: no fibers). Adapted with permission from [66], copyright 2020 Elsevier.

**Figure 4 polymers-14-03333-f004:**
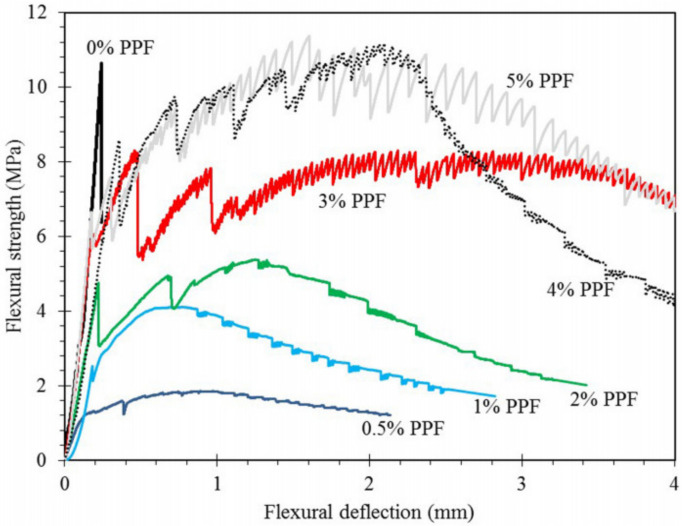
Flexural strength–deflection curve of FOPG at 56 days. Adapted from Ranjbar et al. [88]; licensed under CC BY 4.0 (open access).

**Figure 5 polymers-14-03333-f005:**
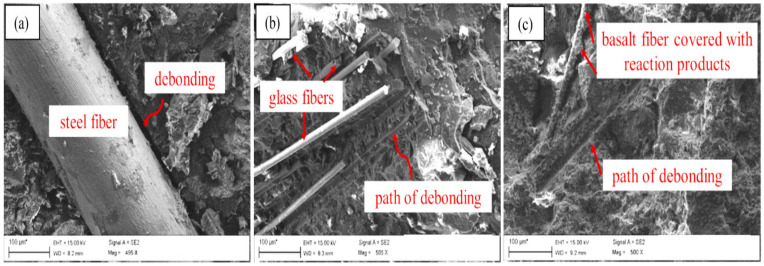
SEM images from the mixture reinforced with: (**a**) steel fiber, (**b**) glass fiber and (**c**) basalt fiber. Adapted with permission from [77], copyright 2021 Elsevier.

**Figure 6 polymers-14-03333-f006:**
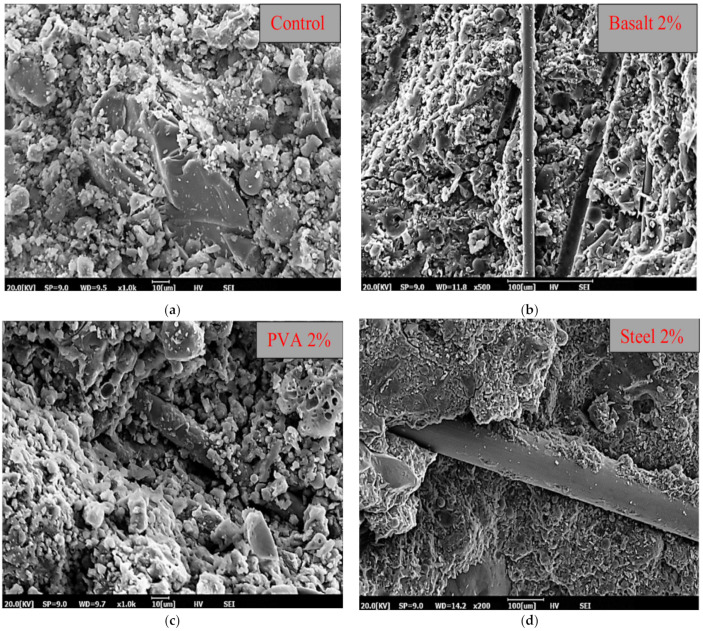
SEM images of FOPGs. (**a**) control, (**b**) basalt fiber, (**c**) PVA fiber and (**d**) steel fiber. Adapted with permission from [66], copyright 2020 Elsevier.

**Figure 7 polymers-14-03333-f007:**
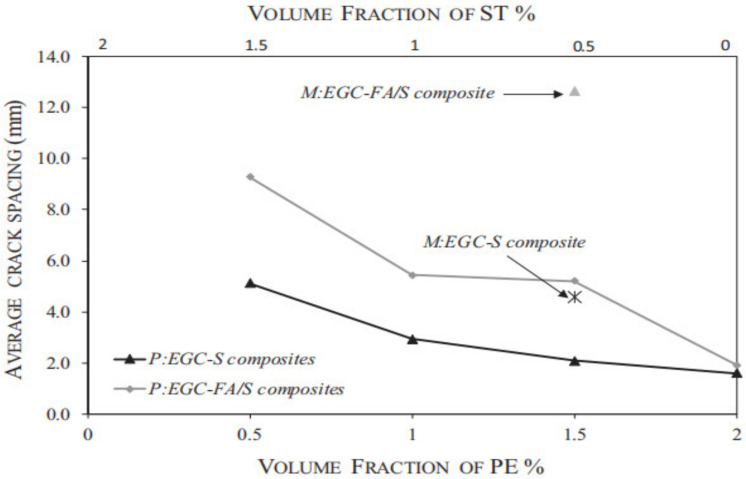
Average crack spacing versus PE and ST fiber volume fractions [69]. Adapted with permission from [69], copyright 2018 Elsevier.

**Figure 8 polymers-14-03333-f008:**
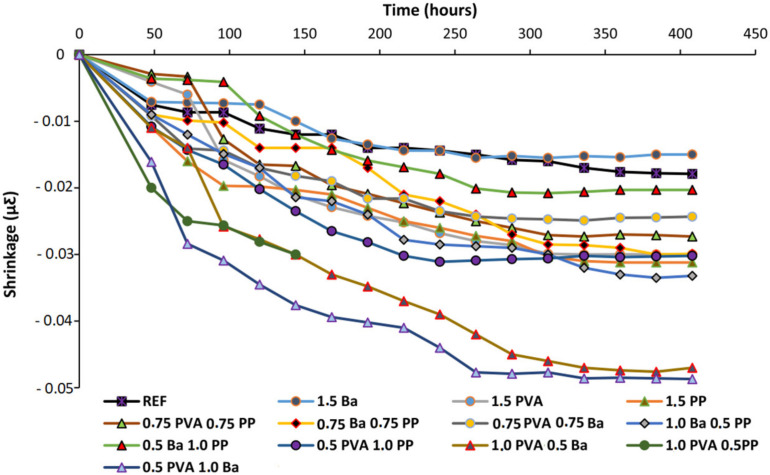
The effects of using different fiber types and combinations on the drying shrinkage. Adapted with permission from [65], copyright 2020 Elsevier.

**Figure 9 polymers-14-03333-f009:**
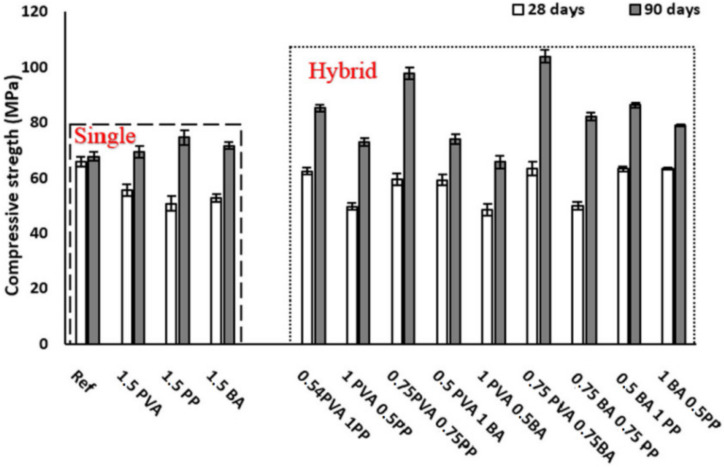
The effects of freeze–thaw cycles on the compressive strength [65]. Adapted with permission from [65], copyright 2020 Elsevier.

**Table 1 polymers-14-03333-t001:** Properties of fibers.

Fiber Type		Aspect Ratio	Elastic Modulus (GPa)	Tensile Strength (MPa)	Density (g/cm^3^)	Elongation (%)	Ref.
Polymer	Polyvinyl alcohol	200	41	1600	1.30	-	[64]
		40–450	-			-	[65]
		308	40			6.42	[66]
		200	41			6	[67]
	Polyethylene	765	114	3000	0.97		[68,69]
		1000	123	3500		3–5	[67]
	Polypropylene	538	-	220–340	0.91	-	[65]
Steel		72	200	2850	7.8	-	[68,69]
		47	200	3000		-	[66]
		47	200	1300		-	[64]
		50	210	850		4.2	[70]
Basalt		333	100	4500	2.63	-	[64]
		1389	-	4100–4840	2.8	-	[65]
		1385	105	3500	2.4	2.5	[66]
		750	45	1360	2.7	1.2	[70]
Carbon		1000	218	3500	1.75	1.5	[70]
Cellulose		117	8.5	750	1.10	-	[64]

**Table 2 polymers-14-03333-t002:** Mix composition of TOPGs.

Precursors	Solid Activators	Fibers	Fiber Content (%)	Fine Aggregate	Curing Condition *	Water Solid Ratio	Ref.
Ground granulated blast furnace slag	Anhydrous sodium metasilicate	Steel	1.0	Standard sand	A, 23 °C 35% (RH)	0.45	[64]
		Polyvinyl alcohol					
		Basalt					
		Cellulose					
Ground granulated blast furnace slag	Anhydrous sodium metasilicate	Polyvinyl alcohol	0.5–1.5	Porcelain ceramic waste	A, 23 °C 60% (RH)	0.35	[65]
		Polypropylene					
fly ash		Basalt					
Fly ash and slag	Anhydrous sodium silicate grains	Micro-steel	0.5–2.0	Fine sand	A, 25 ± 2 °C 70 ± 10% (RH)	0.3	[66]
		Polyvinyl alcohol					
		Basalt					
Fly ash and slag	Anhydrous sodium metasilicate powder	Polyvinyl alcohol	2.0	-	H	0.35	[67]
		Polyethylene			A, 23 ± 3 °C		
Activating slag	Anhydrous sodium metasilicate powder	Steel	1.5 PE and 0.5 ST	Fine silica sand	W	0.45	[68,69]
Fly ash and slag		Polyethylene					
Plain slag	Sodium silicate	Steel	1.0	Standard sand	A, 20 °C 100% (RH)	0.3	[77]
Ternary blended slag		Glass					
		Basalt					
Fly ash and Ground granulated blast furnace slag	Ca(OH)_2_/Na_2_SO_4_	Polyvinyl alcohol	2.0	-	W	0.35–0.375	[78]
	Ca(OH)_2_/Na_2_SiO_3_·5H_2_O				A, 23 ± 3 °C 95 ± 5% (RH)		

* A—Ambient Temperature Air, H—Heat Air, W—Water Tank curing, RH—Relative Humidity.

**Table 3 polymers-14-03333-t003:** Typical coordination ratio and test results of the FOPG. Adapted with permission from [67], copyright 2017 Elsevier.

Mix ID	Fly Ash	Slag	Solid Activator	Water	Fiber	First-Crack Strength (MPa)	Ultimate Tensile Strength (MPa)	Tensile Strain Capacity (%)
PE-H	0.50	0.50	0.08	0.35	PE	2.1 ± 0.24	3.3 ± 0.50	5.5 ± 0.52
PE-A	0.50	0.50	0.08	0.35	PE	2.8 ± 0.66	4.2 ± 0.66	4.9 ± 0.68
PVA-A	0.50	0.50	0.08	0.35	PVA	4.1 ± 0.095	4.6 ± 0.26	4.2 ± 0.71

## Data Availability

Not applicable.

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
