# Peer review of "Properties of Fiber-Reinforced One-Part Geopolymers: A Review"

_polymers, 2022, doi:10.3390/polym14163333_

Round 1
Reviewer 1 Report
The title is interesting regarding fiber-reinforced one-part geopolymers. The review is interesting and relevant to the growing interest in engineering application. However, I feel that, although the authors have put an effort in collecting a lot of information, there is a general lack of consistency and the content of the review is often not sufficiently centered on the main focus of the paper. Hence the manuscript is not acceptable unless substantially revised.
1- Improve the abstract based on the concept of the review.
2- The main massage of the work should be clarify. Please emphasis at the end of introduction.
3- Add a section about fiber-reinforced one-part geopolymer (FOPG)
4- Brief description on figures and table required. Check figures no and also mentioned in the text of the manuscript.
Table 1: please revise the table.
fiber type: it is better use synthetic term for PP, PVA and PE.
Cellulose fibers do not have melting point since they are natural and thermoset.
Diameter or length? Please correct and use the unit.
Figures: please revise the caption for all figures.
Figure 1: Caption should be clarifying. C, B, P, S?
Figure 3 and 4: Use the scale bar in SEM images.
5- Add a paragraph about the limitation of the work and also future perspectives.
6- The main drawback is a substantial lack of technical information on the actual performances of the different kinds of fiber-reinforced geopolymers. While the reader cannot have any idea of the actual findings of the pertinent literature.
In my opinion, the paper will benefit for a thorough revision of the sections, which includes, for example, tables or figures to summarize the findings reported in the pertinent literature for all the different aspect covered in the paper.
7- Improve conclusion part
Author Response
Reviewer 1:
The title is interesting regarding fiber-reinforced one-part geopolymers. The review is interesting and relevant to the growing interest in engineering application. However, I feel that, although the authors have put an effort in collecting a lot of information, there is a general lack of consistency and the content of the review is often not sufficiently centered on the main focus of the paper. Hence the manuscript is not acceptable unless substantially revised.
Thank you very much for reviewing our manuscript in your busy schedule. Based on your valuable suggestions, we have carefully revised our manuscript: Added a table (Table 2, the revised page 5, lines 158-168) and figures (Figure 2 and Figure 4, the revised page 9, lines 337-338; page 11, lines 380-381) on FOPGs related studies, restructuring the sentences and grammatical modification, and a major restructuring of the conclusion. The goal point of this manuscript is achieved successfully. With a distinguished using of very recent references.
1- Improve the abstract based on the concept of the review.
Thank you very much for your review. According to your suggestion, we have carefully enriched the the content of Abstract. (The revised page 1, lines 9-18).
2- The main massage of the work should be clarify. Please emphasis at the end of introduction.
Thank you very much for your review. According to your suggestion, we have carefully revised the section of Introduction. (The revised page 2, lines 57-70)
3- Add a section about fiber-reinforced one-part geopolymer (FOPG)
Thank you very much for your review. According to your suggestion, we have carefully revised the content of this section. (The revised page 3, lines 128-132)
4- Brief description on figures and table required. Check figures no and also mentioned in the text of the manuscript.
Thank you very much for your review. According to your suggestion, we have carefully revised the content of Figures and Tables.
Table 1: please revise the table.
Thank you very much for your review. According to your suggestion, we have carefully revised the content. (The revised pages 3-4, line 150)
fiber type: it is better use synthetic term for PP, PVA and PE.
Thank you very much for your review. According to your suggestion, we have carefully revised the content. (The revised page 4, line 150)
Cellulose fibers do not have melting point since they are natural and thermoset.
Thank you very much for your review. According to your suggestion, we have carefully revised the content. (The revised page 4, line 150)
Diameter or length? Please correct and use the unit.
Thank you very much for your review. According to your suggestion, we have carefully revised the content. (The revised page 4, line 150)
Figures: please revise the caption for all figures.
Thank you very much for your review. According to your suggestion, we have carefully revised the content. (The revised page 3, line 150)
Figure 1: Caption should be clarifying. C, B, P, S?
Thank you very much for your review. According to your suggestion, we have carefully revised the content. (The revised page 10, lines 369-370)
Figure 3 and 4: Use the scale bar in SEM images.
Thank you very much for your review. According to your suggestion, we have carefully revised the content. (The revised page 12, lines 405-406; The revised page 13, line 439)
5- Add a paragraph about the limitation of the work and also future perspectives.
Thank you very much for your review. According to your suggestion, we have carefully revised the content. (The revised page 17, lines 569-587)
- The main drawback is a substantial lack of technical information on the actual performances of the different kinds of fiber-reinforced geopolymers. While the reader cannot have any idea of the actual findings of the pertinent literature.In my opinion, the paper will benefit for a thorough revision of the sections, which includes, for example, tables or figures to summarize the findings reported in the pertinent literature for all the different aspect covered in the paper.
Thank you very much for your review. According to your suggestion, we have carefully rewritten the section in the manuscript.
Table 2 - Mix composition of TOPGs has been added. (The revised page 5, lines 158-168)
Figure 2 - Tensile stress–strain responses of FOPG and Figure 4 - Flexural strength - deflection curve of FOPG at 56 days have been added. (The revised page 9, lines 337-338; The revised page 11, lines 380-381)
7- Improve conclusion part
Thank you very much for your review. According to your suggestion, we have carefully revised conclusion of the manuscript. (The revised page 17, lines 570-589)
In addition, we have also revised other parts of the article according to your review suggestions (highlighted parts in the manuscript).
We are very sorry for the trouble caused to your review due to our bad writing. According to your suggestion, we have carefully and comprehensively revised the manuscript (both in the content and in the language).
Finally, thank you again for your wonderful review of our article in your busy schedule.
Reviewer 2 Report
Authors have reviewed the processability, mechanical and durability properties of FOPG. The paper need an extensive English editing phase because there are many language errors and many sentences have poor meaning. The paper is characterized by many flaws, but after a massive editing (both in the content and in the language) it can be published. In attachment authors can find a PDF file that highlights the main errors in the paper: the highlights without comments indicate an English error that needs revisiting; the highlights with comments, instead, require modifications according the comment.
In addition to the comments in the PDF file, I also have the following suggestions:
1) The number of images in the paper is low: I suggest to add more figures to improve the readability of the paper;
2) All the figures refer to only 3 papers: I suggest to add more figures belonging to a greater number of references;
3) All DOIs are wrong: correct all DOI links.

Author Response
Reviewer 2:
Authors have reviewed the processability, mechanical and durability properties of FOPG. The paper need an extensive English editing phase because there are many language errors and many sentences have poor meaning. The paper is characterized by many flaws, but after a massive editing (both in the content and in the language) it can be published. In attachment authors can find a PDF file that highlights the main errors in the paper: the highlights without comments indicate an English error that needs revisiting; the highlights with comments, instead, require modifications according the comment.
Thank you very much for reviewing our manuscript in your busy schedule. Based on your valuable suggestions, we have carefully revised our manuscript: Added a table (Table 2, the revised page 5, lines 158-168) and figures (Figure 2 and Figure 4, the revised page 9, lines 337-338; page 11, lines 380-381) on FOPGs related studies, restructuring the sentences and grammatical modification, and a major restructuring of the conclusion. The goal point of this manuscript is achieved successfully.
With that sentence the readers can understand that TGP are the only type of geopolymers; rewrite the sentence in order to also include OPG.
Thank you very much for your review. According to your suggestion, we have carefully rewritten the content of Abstract. (The revised page 1, lines 9-18).
I suggest to improve that paragraph because it is not very clear the differences between TPG and OPG: I suggest to start the paragraph with a clear description of the main difference between them.
Thank you very much for your review. According to your suggestion, we have carefully enriched the content of 2. Two-part geopolymer (TPG) and one-part geopolymer (OPG) (The revised page 2, lines 72-73).
I suggest to remove the melting temperature column because it is not relevant for the purpose of the paper and it is pratically empty.
Thank you very much for your review. According to your suggestion, we have remove the melting temperature column (The revised page 4, line 150).
not clear sentence
Thank you very much for your review. According to your suggestion, we have carefully rewritten the sentence. (The revised page 3, lines 147-149).
what authors mean with partial properties?
Thank you very much for your review. According to your suggestion, we have carefully rewritten the sentence. (The revised page 8, lines 319-322).
report here the reference number if the table is reported by another work.
Thank you very much for your review. According to your suggestion, the reference number has been added. (The revised page 8, lines 323).
briefly describe the different curing condition to better understand the difference.
Thank you very much for your review. According to your suggestion, the different curing condition has been added. (The revised page 8, lines 326).
rewrite the sentence, too complex.
Thank you very much for your review. According to your suggestion, we have carefully rewritten the sentence. (The revised page 5, lines 189-190).
explicit all the mechanical properties involved because the "etc" doesn t allow to understand the list of properties.
Thank you very much for your review. According to your suggestion, we have carefully rewritten the sentence. (The revised page 7, lines 236-237).
move this in the figure captions, it is more readable
Thank you very much for your review. According to your suggestion, we have move this in the figure captions. (The revised page 10, lines 369-370).
I suggest to reconsider that sentence: fiber fracture consists in the effective breackage of the fibre during the crack propagation. Please report a reference about that sentence.
Thank you very much for your review. According to your suggestion, we have carefully revised the content. (The revised page 11, lines 391-392)
rewrite that sentence, it is not clear what the authors mean by the term "crack deflection"
Thank you very much for your review. According to your suggestion, we have carefully rewritten the sentence. (The revised page 11, lines 396-398).
rewrite the sentence
Thank you very much for your review. According to your suggestion, we have carefully rewritten the sentence. (The revised page 12, lines 432-435).
redundant phrase
Thank you very much for your review. According to your suggestion, the sentence has been deleted. (The revised page 12, line 439).
Why authors use the term "model"?
Thank you very much for your review. According to your suggestion, we have carefully rewritten the sentence. (The revised page 14, lines 479-480).
Do the acronymus OPG already refer to One Part Geopolymer? why authors repeat "one part"?
Thank you very much for your review. According to your suggestion, we have carefully rewritten the sentence. (The revised page 14, lines 483-485).
"Conductive" is not suitable in that sentence.
Thank you very much for your review. According to your suggestion, we have carefully rewritten the sentence. (The revised page 16, lines 549-550).
In addition to the comments in the PDF file, I also have the following suggestions:
1) The number of images in the paper is low: I suggest to add more figures to improve the readability of the paper;
Thank you very much for your review. According to your suggestion, we have carefully revised figures of the manuscript. (Figure 5, The revised page 12, lines 406-407)
Figure 2 - Tensile stress–strain responses of FOPG and Figure 4 - Flexural strength - deflection curve of FOPG at 56 days have been added. (The revised page 9, lines 337-338; The revised page 11, lines 380-381)
2) All the figures refer to only 3 papers: I suggest to add more figures belonging to a greater number of references;
Thank you very much for your review. According to your suggestion, we have carefully revised figures of the manuscript. we have add more figures belonging to a greater number of references. (The revised page 9, lines 337-338; The revised page 11, lines 380-381; Figure 5, The revised page 12, lines 406-407)
3) All DOIs are wrong: correct all DOI links.
Thank you very much for your review. According to your suggestion, we have corrected all DOI links of the manuscript.
In addition, we have also revised other parts of the article according to your review suggestions (highlighted parts in the manuscript).
We are very sorry for the trouble caused to your review due to our bad writing. According to your suggestion, we have carefully and comprehensively revised the manuscript (both in the content and in the language).
Finally, thank you again for your wonderful review of our article in your busy schedule.
Round 2
Reviewer 1 Report
The authors have done a nice job for response the reviewer comments. Hence still some concerns that should be addressed:
1- Add a paragraph about the limitation of the work and also a section for future perspectives.
2- Improve conclusion part, do not use multi-paragraph style for conclusion.
Author Response
The authors have done a nice job for response the reviewer comments. Hence still some concerns that should be addressed:
1- Add a paragraph about the limitation of the work and also a section for future perspectives.
Thank you very much for your review. According to your suggestion, we have carefully revised the content. (The revised page 17, lines 548-571)
2- Improve conclusion part, do not use multi-paragraph style for conclusion.
Thank you very much for your review. According to your suggestion, we have carefully revised the conclusion. (The revised page 17, lines 572-591)
Finally, thank you again for your wonderful review of our article in your busy schedule.